# RAMCOM: A qualitative study of clinicians' viewpoints on a tool for communication with Muslim patients considering fasting during Ramadan

**Mohamed Ezzat Khamis Amin**[1]*, **Ahmed Abdelmageed**[2]

**1** Department of Pharmacy Practice, Faculty of Pharmacy, Beirut Arab University, Beirut, Lebanon,
**2** Department of Pharmacy Practice, Manchester University College of Pharmacy, Fort Wayne, Indiana, United States of America

* m.amin@bau.edu.lb

**Data Availability Statement:** Data cannot be shared publicly because the study analyzes qualitative data and the participants did not consent to have their full transcripts made publicly

## Abstract

### Objectives

Fasting during the month of Ramadan is a basic pillar of Islam. While patients may be religiously exempted from fasting, literature indicates that the majority decide to fast. Caring for millions of Muslim patients who decide to fast during Ramadan can be challenging for clinicians around the globe. This study proposes a communication tool, RAMCOM, which aims to assist clinicians in communicating with Muslim patients considering fasting during Ramadan. It addresses the following questions: What are the clinicians' preferences for the tool in terms of Content, Format, Style, Length and language? How do clinicians perceive factors that would impact their intentions to sue the tool? What would facilitate the implementation and dissemination of the proposed tool?

### Methods

Semi-structured interviews were conducted with a purposeful sample of clinicians in Egypt (11) and the US (10). Clinicians were purposefully sampled to assure variance in age, gender, time in practice, specialty, and religious background. Directed content analysis was conducted and emerging data were mapped to constructs within the theory of planned behavior. Iterative sampling and analysis continued until saturation was reached.

### Results

In total, 21 clinicians were interviewed. The tool was iteratively revised according to clinicians' comments on format, content, language and usability. Factors contributing to using RAMCOM included perception of tool (need for use, perceived burden of use), perceived norm (perceived patient expectations), and ability to use tool (time, frequency of seeing patients, knowledge of Ramadan and Islam). Practice environment factors that impact the use of RAMCOM include education, early reminders, colored laminated copies, communication training, involvement of support staff, and patient education.

available. If requested, excerpts of the transcripts relevant to the study would be shared. Requests for excerpts may be sent to the corresponding author at m.amin@bau.edu.lb Alternatively, requests can be made to Dr. Robert D. Beckett (rdbeckett@manchester.edu), IRB Chair and Fort Wayne Campus Lead at Manchester University

**Funding:** The author(s) received no specific funding for this work.

**Competing interests:** The authors have declared that no competing interests exist.

## Conclusion

Clinicians provided valuable perceptions on the implementability and use of RAMCOM, a new communication tool designed to assist in caring for Muslim patients during Ramadan. These perceptions should be considered by different stakeholders to facilitate goal-concordant care for Muslim patients considering fasting.

## Introduction

Ramadan, the ninth month in the lunar calendar, is a sacred month when hundreds of millions of Muslims refrain from eating, drinking and marital sex from dawn until sunset. This is a religious obligation for able-bodied adult Muslims, which has been prescribed in the Qur'an 2:183–184 [1]. It's an opportunity for a person to train to exercise self-restraint and reflect. This month is filled with celebration and spirituality as Muslims strive to read the Qur'an and invest their time in reflecting and prayers.

Islamic Jurists have had differences in opinions on the kinds of dosage forms that nullify fasting[2]. Hence, a religious-medical symposium was held to address differences on dosage forms that nullify fasting [3]. It concluded that dosage forms such as oral medications, eye and ear drops, topical preparations, suppositories and pessaries, injections, nitroglycerine sublingual tablets as well as mouth washes and oral sprays do not nullify fasting. The symposium's participants, however, could not reach a consensus on nose drops, nose sprays, anal injections and inhalers. Important to note is that the plurality of opinions (ikhtilaf) within jurists is usually seen in positive light in Islam, As Al-Imam Al-Shafi'i said, "I believe my opinion is right with the possibility that it is wrong. And I believe the opinion of another who disagrees with me is wrong with the possibility that it is right" [4].

Intent plays an important role in determining if an action invalidates fasting [5]. As the Qur'an 2:183–184 requires Muslims to fast [1], it's Haram (prohibited) for Muslims not meeting any exemptions to eat or drink in the period from dawn to sunset during Ramadan. However, according to the Hadith (record of the traditions or sayings of the Prophet Mohammad), if a fasting person ate or drank out of forgetfulness during fasting hours, his/her fast remains valid [2]. Thus, the use of oral medications, which is normally intentional, nullifies the fast. This opinion represents the majority of Islamic scholars [3]. Accordingly, such medications can only be used in the period between sunset and dawn when consuming them does not nullify fasting. Out of ease for worshippers, Islam's teachings provide an exemption for individuals meeting certain conditions as demonstrated by the Qur'an 2:185 [1]. Those include Muslims who are sick or traveling for more than 81 Km (50 miles)[1]. The Egyptian Dar Al Ifta, a Sunni resource, recommends that those individuals must make up for all missed days upon recovery, return, or after weaning[6]. Scholars had different opinions concerning the ruling of what pregnant woman and breastfeeding mothers should do if they do not fast [7]. Those opinions ranged from them having to make up the fasts only, make up the fasts and feed one poor person for each day if they fear for their children, or feed the poor only without having to make up the fasts.

Senior Muslim men and women who find it difficult to fast as well as patients with chronic conditions whose health may be negatively influenced by a change in medication use or diet associated with fasting are exempt [3]. In fact, they should not be participating in the ritual to prevent self-harm, which is prohibited according to the Qur'an 2:195 "and let not your own hands throw you into destruction."[1] While Muslim patients need not apply for an exemption from a religious authority, literature indicates that some Muslim patients like to "consult

imams, religious scholars and clinicians when trying to identify whether they have a religious exemption from fasting or not and which dosage forms may or may not nullify it" [8,9]. This is particularly significant since some differences may exist about the interpretation of exemptions from fasting within and between sects [10,11]. Moreover, patients have different views in relation to dosage forms that can be used from dawn to sunset while fasting. For instance, some patients believe that the use of dosage forms such as asthma devices, drops, rectal suppositories, and injections would nullify their fasting while other patients believe that some or all of those dosage forms can be used without impacting fasting [12–14]. A large study published in 2004 included thousands of Muslim patients in thirteen countries and showed that nearly eight out of ten patients with type 2 diabetes chose to fast although they might have been aware of their religious exemption from fasting [15]. Similar results were reported in a recent study that has been carried out in six countries with the majority of diabetic patients deciding to fast [16]. In some cases, patients may change their therapeutic regimens without consulting a clinician resulting in patients omitting doses, taking medications at intervals that are too close, or at times stopping one or more medications entirely, which could lead to increased risks for those patients [9,17].

A number of studies, the majority of which were done in non-Muslim countries, addressed the communication lapses between clinicians and patients who decide to fast during Ramadan [18–27]. At times, the topic is never addressed in clinical encounters that occur before or during Ramadan. The topic may be addressed in an unsatisfactory fashion leaving key information such as medication regimen adjustment, potential risks of fasting, signs indicating the need to break fasting without being addressed [14].

While some guidance has been provided to clinicians when communicating with patients about Ramadan, [28,29] there is no structured published tool for clinicians to use when communicating with patients who are considering fasting during Ramadan. This manuscript is part of a larger study dataset examining clinician-patient communication in relation to Ramadan. The first manuscript used social cognitive theory as a guiding framework to explore factors affecting counseling provided to Muslim patients considering fasting Ramadan from a clinician's perspective [14]. This second manuscript complements earlier findings by proposing a communication tool, RAMCOM, which aims to assist clinicians in communicating with Muslim patients considering fasting during Ramadan. Using a theory driven qualitative study design, this study addresses the following questions: What are the clinicians' preferences for the tool in terms of Content, Format, Style, Length and language? How do clinicians perceive factors that would impact their intentions to sue the tool? What would facilitate the implementation and dissemination of the proposed tool?

## Methods

What follows is a description of methods used, which like the rest of the manuscript, adheres to Standards for Reporting Qualitative Research (SRQR)[30].

### Construction of initial draft of RAMCOM

This tool was designed with the goal of providing clinicians with a resource that would assist them in providing patient-centered care to Muslim patients who are considering fasting during Ramadan, hence the name RAMCOM standing for RAMadan COMmunication, To achieve this, the proposed tool follows the seven essential sets of communication tasks described in the Kalamazoo consensus statement: (1) build the doctor-patient relationship; (2) open the discussion; (3) gather information; (4) understand the patient's perspective; (5) share information; (6) reach agreement on problems and plans; and (7) provide closure.[31]

Moreover, specific components of RAMCOM address issues that have been identified from research related to clinician patient communication about Ramadan. Earlier research indicates that patients generally preferred more autonomous decision making when it comes to fasting during Ramadan. Accordingly, the tool follows a model of shared decision making where both the patient and the clinician contribute to the decision. It also helps clinicians show some awareness of the basic tenets of Ramadan fasting and its significance early in the encounter. It encourages clinicians to demonstrate empathy whenever an opportunity is provided, which has been shown to improve the quality of communication and care, especially in cross cultural communication [32].

The tool encourages clinicians to ask patients whether they think the dosage form(s) they are currently using nullifies their fast or not. If a patient thinks the dosage form nullifies their fast, then it cannot be taken from dawn to sunset while fasting, but if possible, it might be moved to the period from sunset to dawn. In case the patient decides to fast, the tool encourages clinicians to be explicit about potential risks of fasting, use of medications, indications to break fasting, diet and meal times, and exercise. The use of written materials when possible is also emphasized.

Given that the tool was introduced to Arabic speaking clinicians in Egypt along with English speaking clinicians in the United States, English and Arabic versions of RAMCOM were prepared by the first author. To ensure consistency among the Arabic and English versions of the tool, an independent physician who is fluent in both languages was recruited to verify the accuracy of the translation.

## Participants and recruitment

Ethical approval for the study was obtained from the Manchester University Institutional Review Board with written informed consent from participants (Study Number: 1617FW12). Recruitment and interviews of clinicians took place from May to September 2017. In 2017, Ramadan (Hijri year 1438) started on May 26[th] and ended on June 24[th]. Details of sample selection, recruitment and strategies addressing trustworthiness as defined by Lincoln and Guba [33] have been described in the first manuscript published from this project [14]. Ongoing preliminary data analysis has been conducted during data collection. The study did not have a target sample size; rather it aimed to recruit participants until saturation of key themes occurred. More specifically, a strategy involving a priori thematic saturation, which relates to the degree to which identified codes or themes are exemplified in the data, was followed [34]. The process of iterative sampling and analysis continued throughout the study.

## Interview method and research instrument

Interviews were conducted in-person (11) and over the phone (9). All interviews were conducted by the first author with the exception of one, which was conducted by the second author. All interviews were recorded using an audio recorder with participant permission. Recordings were transcribed verbatim following each interview by native speakers in the language used to conduct the interview. Field notes were created by the first author during interviews. These notes were used to supplement the interview data and helped the research team in producing meaning and understanding of the issue being studied.

The most recent version of the tool was sent electronically to each clinician ahead of each interview. The interview began in an open-ended manner, getting participants to talk about their typical encounter with Muslim patients before Ramadan. Open-ended, neutral questions were carefully constructed to reduce the chance of eliciting socially desirable responses. The interviewer asked participants to reflect on any factors that would facilitate or prohibit them

from using the tool. Items derived from the GUIDE-IT tool framed questions of the initial interview guide that addressed the tool's implementability [35]. Participants were also asked about how they think the tool should be introduced to clinicians and how to have it available at their practice sites. Appropriate probes were used throughout the interviews when the interviewer wanted to obtain a more in-depth response. To establish face validity, the interview guide was pre-tested with an Egyptian physician. Only minor changes to how the researcher phrased the questions resulted from pre-testing.

## Data analysis

The main unit of analysis was the individual narratives of interviewed clinicians. The coding process began with the identification of passages from the transcribed interviews which fit the constructs, yielding to the reported themes. The first author started familiarizing himself with the data through reading and re-reading transcripts, noting the participants' use of particular words or thought patterns emerging from the data. This was followed by discussions among the two authors. The template analysis style where the text was organized and presented according to preexisting categories was used[36]. Directed content analysis[37] was performed based on components of the GUIDE-IT tool[35] including format, content and language and the theory of planned behavior (TPB)[38]. According to the TPB, attitudes (clinicians' attitude towards RAMCOM), subjective norms (perception of whether patients approve using RAM-COM) and perceived behavioral control (perception of the difficulty of using RAMCOM in practice) determine the individual's behavioral intention (clinicians' plan to use RAMCOM in caring for patients) and consequently determine the likelihood of the individual carrying out that specific behavior.

## Results

In total, 20 physicians and one ambulatory care pharmacist were interviewed. Participants were practicing in Egypt (11) and the US (10); thirteen were male. Experience as practitioner ranged from 11 months to 38 years. Nine participants were non-Muslims including two non-Muslim clinicians from Egypt and seven from the US. Self-identified clinicians' areas of practice included Family Medicine (6), Internal Medicine (3), Diabetology (3), Tropical Medicine & Hygiene (2), Pulmonology (1), Cardiology (1), Gastroenterology (1), Ophthalmology (1), Nephrology (1), Psychiatry (1), and Obstetrics & Gynecology (1). Participants' average age was 45 and ranged from 28 to 68 years. Two participants did not read the tool prior to the interview and were unable to provide comments on its content and usability. They did, however, provide comments on dissemination and other issues related to barriers and facilitators for implementation. The mean duration for those interviews was 30 minutes (range 21–44).

## Perceptions of RAMCOM

**Content, format, style, length and language.** Several clinicians preferred the tool to be only one page long. Accordingly, the first draft of the tool, which was nearly 2–3 pages long was shortened to one page. The final version of RAMCOM is shown in Fig 1.

There was considerable variation in preference for hard copy versus electronic versions of the tool with some clinicians preferring to have both versions available. Specifically, clinicians cited the possibility of giving a colored laminated copy of the tool to the patient during discussions to build rapport as reasons for preferring a hard copy version of the tool. Seeing the entire process on a single piece of paper rather than having to go through multiple screens in an electronic version was an additional reason for preference. Others clinicians, however, preferred having an electronic version citing the ability to pull it on a smart phone anywhere and

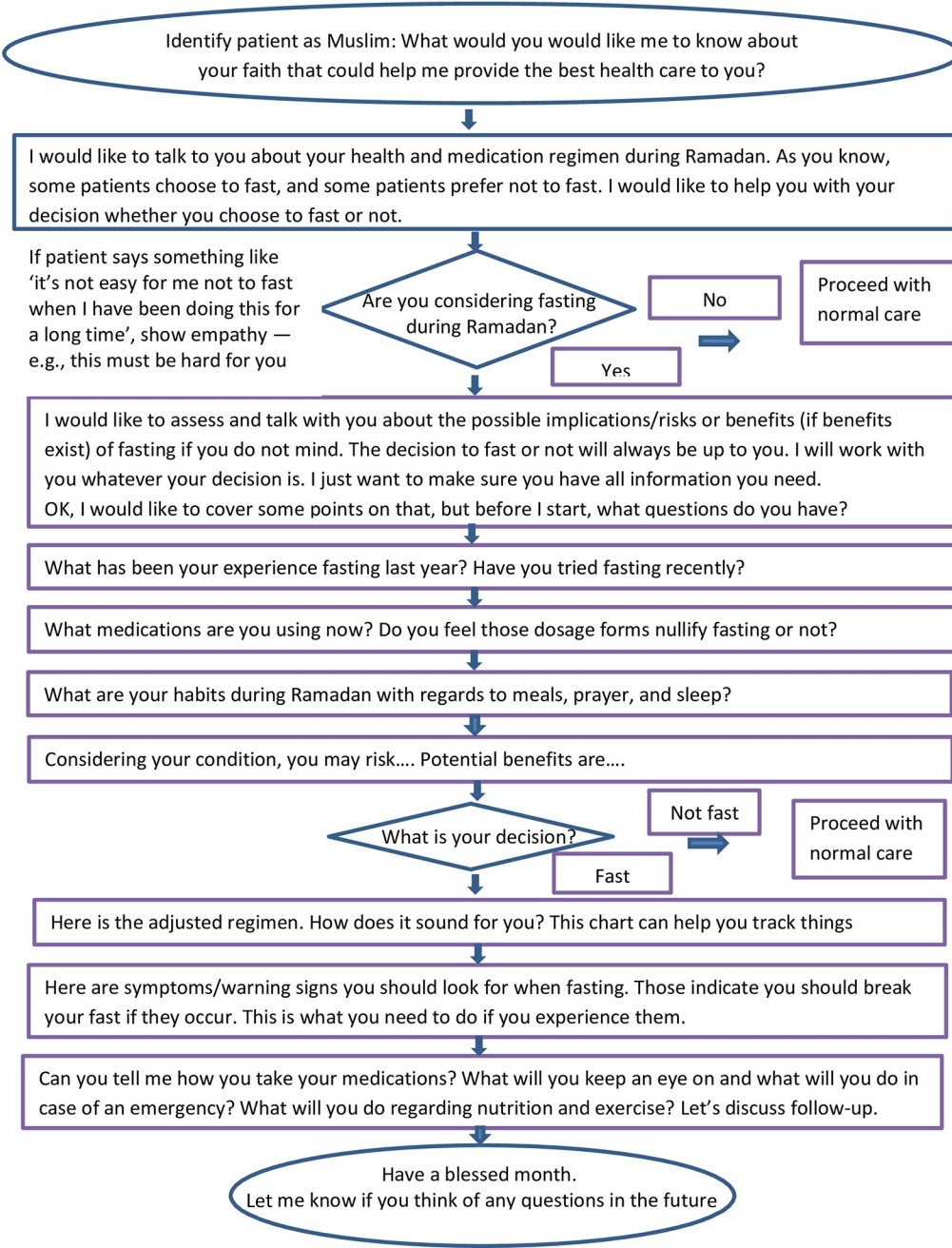

**Fig 1. RAMCOM communication tool.**

at anytime as compared to a hard copy tool that could be set aside and lost. The ability to access other information simultaneously through pulling another screen was viewed as an additional advantage. Egyptian clinicians, however, cited the inaccessibility of technology in many settings as a significant barrier.

While most clinicians felt that they would prefer the flowchart format given the way it is organized and its similarity to other algorithms they use, one clinician felt the bullet point format provides better flexibility that would help him better adapt the interview to the patient.

Participants' suggestions about the language used were minimal and were addressed as the study proceeded. A clinician in Egypt indicated that he particularly appreciated that the Arabic version of the tool was prepared in the Egyptian dialect, which would help clinicians convey messages clearly to patients without the burden of translating it from English or classical Arabic. Several clinicians indicated they preferred the tool in the form of scripted questions and phrases to say rather than directions of what should be done. The tool was revised in accordance with this format.

**Value of tool.** Several US clinicians felt that parts of the tool, such as the one where they ask patients on whether they feel a dosage form would nullify fasting or not, were informative in themselves. While some clinicians felt that the tool would be useful for them to utilize, others, showing signs of high self-efficacy for caring for patients in relation to Ramadan, felt they don't need to use it because dealing with patients who observe Ramadan on a regular basis has given them the necessary skills to drive such conversation. Negative prior perceptions about other regimented tools and policies regarding their use, especially with US clinicians who frequently deal with them, were cited as a reason why clinicians may resist using RAMCOM. They specifically cited earlier exposures to other tools where they had to provide evidence of addressing each item in the electronic system. Physician A explains:

> "When we have to go through questions on a computer, sit down, and go through a checklist to make sure we are knowledgeable of this kind of thing, or this new policy, we hate it. And I will tell you we go through it fast enough just to say that we did it to be able to sign off."

**Perceived norm.** A key issue for clinicians was on how to proceed with the first step of the tool, which involves identifying the patient as a Muslim. While Egyptian clinicians, including non-Muslim ones, did not cite this as a problem, perceptions of US clinicians were mixed. Some US clinicians felt that they were comfortable addressing this, especially if the patient has a Muslim name or wear hijab. Others felt that the norm in the US makes it uncomfortable both for clinicians and patients to bring this up.

Three Egyptian clinicians argued that they would reserve the concordant approach of using the tool for patients who would be a good fit for this kind of shared decision making. When probed about the criteria they use to identify such patients, they described patients with higher degree of education. Physician N explains how using a concordant approach with patients of lower education would defy patients' expectations for his role and would result in patients having doubts about his ability to manage their condition:

> "A patient may not like this way (of communication). Some patients like to have the doctor to be their superior *'Do this and take that treatment. Don't discuss it with me'*. That's the majority, especially those, I am sorry to say, of low socioeconomic status. If I do this (use the tool), they would say *'the doctor is not good, he keeps on asking us questions. What kind of a doctor would keep on asking me questions? Why am I coming to see him then'. . ..* and they would leave me"

**Ability to use tool.** Awareness about Islam and Ramadan was a key factor impacting the clinicians' ability to use the tool. Non-Muslim clinicians in the US, especially, ones who had limited to no prior knowing exposure to Islam and Muslims felt that cultural competency training would help them in using the tool more effectively to address patients' needs. Time spent with patients was cited as another significant factor that would impact the clinician's ability to

use the tool. As expected, clinicians who feel that they have time to address the issue of fasting thoroughly were more confident about their ability to use RAMCOM as a guiding tool.

**Clinicians' suggestions for dissemination and implementation.** Clinicians recommended several strategies for disseminating the tool. Those included covering the tool in debriefings and one to one clinician visits and introducing it at national and international conferences. Younger clinicians suggested incorporating the tool in cell-phone applications and adding it to electronic evidence-based clinical decision support resources such as UpToDate. Involving different stakeholders including medical societies, universities and government agencies was cited as a strategy that would help in disseminating the tool and sustaining its use.

Training and educating clinicians and their support staff on issues related to Ramadan, Islam and communication with patients in general were key recommendations for facilitating the use of RAMCOM in practice. The use of case studies in training was specifically cited as a potentially helpful strategy. Should they be available, coupling RAMCOM with clinical guidelines for different conditions was also viewed as something that would help clinicians utilize it. Given the fact that the lunar calendar does not correspond with the ubiquitously used Gregorian calendar, Ramadan occurs around 11 days earlier every year. Accordingly, clinicians felt that sending annual reminders about the tool and its use weeks to months before Ramadan could be a useful strategy for institutions. Identifying and making outreach efforts to Muslim patients in the community and patients served by clinicians' institutions was viewed as an additional strategy that could help patients bring the topic to clinicians, which would enable them to identify patients to go over the tool with.

## Discussion

Building on the literature on implementability, [35] this study describes the process of developing RAMCOM, a communication tool designed to assist clinicians in communicating with patients about fasting during Ramadan. As envisioned, examining constructs related to perceived value, subjective norm and perceived ability to use the tool provided useful insights in relation to the tool's implementability. In addition, the study provided guidance from clinicians in different settings on useful strategies for disseminating the tool.

Findings from this study have implications when considering the specific context in which the tool will be used. As indicated by clinicians, preferences can vary considerably not just in different countries and health systems, but even within a single institution. For instance, while opportunities for integration of the tool within electronic resources should be explored, it is important to provide clinicians with preference for hard copies of RAMCOM an opportunity to apply it in their practice settings. Likewise, dissemination attempts should be adapted to address audience in different specialties and geographic places rather than solely focus on clinicians dealing with diabetic patients.

Moreover, for a better use of the tool, patients should be better empowered to ask questions. Training patients to take the initiative and start a conversation with clinicians on the topic would make it easier for clinicians to apply the tool. Literature, however, indicated that patients are reluctant to bring this up for a number of reasons, including prior experiences with clinicians and patient concerns about straining their relationship with clinicians and perceptions of Muslims among clinicians with little background of dealing with this group [23,39]. One way of empowering patients to address this issue is to provide them with a patient version of the RAMCOM tool that mirrors the clinician version of the tool [9]. This could increase the chances of having a more goal-concordant care for this class of patients.

Respecting the autonomy of patients with different backgrounds is a key principle of biomedical ethics. Some clinicians in this study, however, indicated they were unlikely to use the

tool as part of a concordant approach with low socioeconomic patients. These results corroborate the findings of a great deal of the previous work in clinician patient communication, which indicates that the doctors commonly misperceive the desire and need for information of patients of low socioeconomic status along with their ability to contribute to making decisions in their care process. More specifically, patients with low socioeconomic status receive "less positive socio-emotional utterances and a more directive and less participatory consulting style, characterized by significantly less information giving, less directions and less socio-emotional and partnership building utterances from their doctor" [40]. An alternative approach would involve assessing preferences for patients with all backgrounds, rather than only ones with low socioeconomic status, on whether they wish to participate in decisions or whether they wish to delegate them to clinicians. This should be done while noting that a patient's role preference might change as the visit proceeds [41].

It is important to note that, ideally, such a tool should be viewed as one of several components in interventions addressing this issue. To be successful, a comprehensive intervention would involve all aspects impacting the patient care process in relation to Ramadan. One strategy for this is to use RAMCOM along with AIDET, a communication framework that has been shown to improve patient satisfaction and increase the ability of the patient to participate in their care [42,43]. The acronym AIDET® stands for five communication behaviors: Acknowledge, Introduce, Duration, Explanation, and Thank You [44]. Another helpful strategy would be to compile a list of available religious teachings and resources sorted by sect and/or madhab addressing this topic that could be listed on the back side of the hard copy of RAMCOM for further reference for patients or providers. Moreover, institutional support is extremely important in overcoming organizational barriers in implementing such initiatives. It helps in making the healthcare facility a Muslim friendly and welcoming place to Muslim patients as well as providing clinicians with proper cultural competency training addressing Muslim patients, Ramadan, and other background information that would help in applying the tool.

At this point in time, clinical guidelines for transitioning patients safely in and out of Ramadan are generally lacking. There are, however, some intensive resources for therapeutic guidance on caring for diabetic patients produced by the Diabetes and Ramadan International Alliance (DaR) and other collaborators such as Diabetes Canada [45]. Those guidelines can be used along with RAMCOM when training clinicians to assist them in the care process. Sessions introducing the patient's perspective could be helpful, especially in cases where clinicians are under the false assumption that patients are satisfied with the communication process when in fact, they may not be. Finally, RAMCOM should be introduced to clinicians as a guiding tool that would help them in the care process rather than an additional tool that is imposed on them, to avoid triggering the resistance clinicians have reported with regimented tools previously introduced at their practice sites.

## Strengths and limitations

This study sample included clinicians from countries with different proportions of Muslim populations and different health systems. This in addition to the considerable variation in characteristics and perspectives of participating clinicians brought richness to generated data addressing RAMCOM, its implementability and dissemination. On the other hand, it is possible that some clinicians refrained from giving negative criticism about the communication tool knowing that the authors have designed it. It is likely that discussions following wider dissemination of the tool would further help in refining it.

## Conclusions

Clinicians provided valuable perceptions on the implementability of RAMCOM and its use in caring for Muslim patients. To facilitate better care for Muslim patients considering fasting, these perceptions should be considered by different stakeholders. Further studies should explore the value of RAMCOM in future interventions addressing this communication gap. More specifically, follow-up studies should address the implementation and refinement of RAMCOM to see its effectiveness in different settings including clinicians from other specialties such as emergency medicine physicians, while offering Muslim patients a version of RAMCOM they can use from their end.

## Acknowledgments

We are very indebted to colleagues who supported this project in Egypt and the US. We are particularly thankful for Dr. Betty Chewning for valuable methodological advice, Dr. Ahmed Osman, Dr. Mohammad Shoukry Newegy, Dr. Brian Henriksen, and Dr. Joshua Kline, for technical support in the collection of data, Mrs. Eman ElNaggar, Ms. Marwa Farhat for transcription of data. We would also like to thank all clinicians who took the time to participate and provide valuable perceptions about the tool in return of a small monetary compensation.

## Author Contributions

**Conceptualization:** Mohamed Ezzat Khamis Amin, Ahmed Abdelmageed.

**Data curation:** Mohamed Ezzat Khamis Amin, Ahmed Abdelmageed.

**Formal analysis:** Mohamed Ezzat Khamis Amin, Ahmed Abdelmageed.

**Investigation:** Mohamed Ezzat Khamis Amin, Ahmed Abdelmageed.

**Methodology:** Mohamed Ezzat Khamis Amin, Ahmed Abdelmageed.

**Project administration:** Mohamed Ezzat Khamis Amin, Ahmed Abdelmageed.

**Resources:** Mohamed Ezzat Khamis Amin.

**Supervision:** Mohamed Ezzat Khamis Amin.

**Writing – original draft:** Mohamed Ezzat Khamis Amin, Ahmed Abdelmageed.

**Writing – review & editing:** Mohamed Ezzat Khamis Amin, Ahmed Abdelmageed.

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
