## [Decision Letter · Decision Letter 0]

2 Dec 2019

PONE-D-19-30447

RAMCOM: A tool for communication with Muslim patients considering fasting during Ramadan

PLOS ONE

Dear Dr. Amin,

Thank you for submitting your manuscript to PLOS ONE. After careful consideration, we feel that it has merit but does not fully meet PLOS ONE’s publication criteria as it currently stands. Therefore, we invite you to submit a revised version of the manuscript that addresses the points raised during the review process.

We would appreciate receiving your revised manuscript by Jan 16 2020 11:59PM. To enhance the reproducibility of your results, we recommend that if applicable you deposit your laboratory protocols in protocols.io, where a protocol can be assigned its own identifier (DOI) such that it can be cited independently in the future. For instructions see: http://journals.plos.org/plosone/s/submission-guidelines#loc-laboratory-protocols

We look forward to receiving your revised manuscript.

Kind regards,

Andrew Carl Miller, M.D.

Academic Editor

PLOS ONE

Journal Requirements:

1. We noticed you have some minor occurrence of overlapping text with the following previous publication(s), which needs to be addressed:

https://link.springer.com/article/10.1007%2Fs10943-019-00820-y

The text that needs to be addressed involves parts of the Introduction.

In your revision ensure you cite all your sources (including your own works), and quote or rephrase any duplicated text outside the methods section. Further consideration is dependent on these concerns being addressed.

Additional Editor Comments (if provided):

Thank you for the opportunity to review this well written and important article. Please see the reviewer comments for suggested revisions.

Reviewers' comments:

Reviewer's Responses to Questions

**Comments to the Author**

1. Is the manuscript technically sound, and do the data support the conclusions?

Reviewer #1: Yes

Reviewer #2: Yes

Reviewer #3: Partly

2. Has the statistical analysis been performed appropriately and rigorously? 

Reviewer #1: N/A

Reviewer #2: N/A

Reviewer #3: N/A

3. Have the authors made all data underlying the findings in their manuscript fully available?

Reviewer #1: No

Reviewer #2: Yes

Reviewer #3: No

4. Is the manuscript presented in an intelligible fashion and written in standard English?

Reviewer #1: Yes

Reviewer #2: Yes

Reviewer #3: Yes

5. Review Comments to the Author

Reviewer #1: The paper is well written and described the instrument and the clinicians perspectives of it. To allow future use and comparison studies, please provide the Egyptian dialect version of Arabic as a supplementary material

Reviewer #2: This is an excellent innovative study where 21 Muslim and non-Muslim clinicians provided valuable feedback on the implementability and use of RAMCOM, a new communication tool designed to assist in caring for fasting Muslim patients. I have revised the whole manuscript and RAMCOM tool (figure 1) and added my questions and comments in the attached word document. There are also few minor grammatical errors. Please send me the final draft after you answer my few questions and edit the manuscript for a final revision. Again, thank you for the well written and interesting article. I am sure it will have a great benefit to Muslim patients around the world.

Best regards,

Heba Abolaban, MD, MPH

Reviewer #3: Thank you for the opportunity to review this manuscript exploring an important healthcare need affecting Muslim patients.

General

Please be sure the manuscript adheres to & cites the appropriate Equator Network guidelines. For example: O'Brien BC, Harris IB, Beckman TJ, Reed DA, Cook DA. Standards for reporting qualitative research: a synthesis of recommendations. Acad Med. 2014;89(9):1245-1251.

TITLE

Please provide a concise description of the nature and topic of the study Identifying the study as qualitative or indicating the approach (e.g., ethnography, grounded theory) or data collection methods (e.g., interview, focus group)

ABSTRACT

Please clarify any hypothesis tested.

INTRODUCTION

Please provide a more referenced review of relevant theory and empirical work. This may include references to the Qu’ran, Sunnah (in form of Hadith), or ijtihad (independent legal reasoning), istihsan (preferential reasoning of jurists), al-urf (local customary precedent), and al-masalih al-mursalah (public interest or welfare) among others. There is a juristic heirarchy and these do not all rise to the same levels of precedent. Also consider making the distinction between Halal (permissible or lawful), Haram (prohibited), and Makruh (discouraged but not legally forbidden).

Also important to note is the idea that under Islamic law “ijtihad is not reversible” (al-ijtihad la yunqad), meaning that one ruling of ijtihad is not reversed by another of differing opinions. Disagreements (ikhtilaf) among jurists are seen in a positive light; legal texts record different juristic opinions on the same issue with a specific line of literature devoted to disagreements between jurists (ikhtilaf al-fuqaha). This juristic ikhtilaf is key to understanding the development of the Islamic legal tradition, and can provide an important juristic tool to interpreting Sharīʿah Law as it pertains to health and medicine. Many find this idea confusing. Indeed, classical Muslim scholars have reminded us that: our opinion is a right one with the possibility of being wrong and others’ opinions are wrong ones with the possibility of being right.

Line 59-60: Consider further clarifying the root of this obligation. For example: It is a religious obligation for able-bodied adult Muslims as prescribed in the Qur’an 2:186. Fasting is obligatory upon every adult Muslim who is of sound mind, not ill, and not travelling > 80 km from city of residence (Quran 2:184).

Line 61-62: Intent plays an important role in determining if an action invalidates ones fast. Consider clarifying this. For example: Many things that come into or out of the body invalidate fasting, such as intentional eating or drinking, oral medications that reach the stomach, deliberate vomiting, parenteral nutrition, and others.

Lines 65-58: Please provide more references on these topics as they are important and there are numerous fatawat on the topic; I know of at least 13 (1 Sunni, 11 Shi’a, 1 joint Sunni and Shi’a). When citing any fatawa, please be sure to specify if it is Sunni or Shi’a, and the Madhab if known.

Line 72-74: This may be justified by Qur’an 2:195: …and let not your own hands throw you into destruction.

Please clarify any hypothesis tested.

METHODS

Lines 182-185: Might consider incorporating emergency medicine physicians on future studies, as they may play a unique role when evaluating patients during more acute illnesses.

RESULTS / DISCUSSION

Thank you for including a copy of the tool. One might consider compiling a list of available evidence sorted by sect and/or madhab addressing this topic that could be listed on the back side of the page for further reference for patients or providers.

6. PLOS authors have the option to publish the peer review history of their article (what does this mean?). If published, this will include your full peer review and any attached files.

Reviewer #1: No

Reviewer #2: Yes: Heba Abolaban

Reviewer #3: Yes: Andrew C. Miller M.D.

---

## [Author Response · Author response to Decision Letter 0]

17 Jan 2020

Andrew Carl Miller, M.D.

Academic Editor

PLOS ONE

Dear Dr Miller,

 It is with pleasure that we submit to you a revised version of manuscript PONE-D-19-30447. We genuinely appreciate the insightful comments that you and the reviewers have provided. We have incorporated suggestions, which we believe strengthened the manuscript. We thank you and the reviewers for your work on this, for your encouraging words and for noting the importance of addressing this topic. We list manuscript revisions and response to comments below.

REVIEWER COMMENTS

Editor:

• Thank you for the opportunity to review this well written and important article. Please see the reviewer comments for suggested revisions.

• Thank you for your encouraging words and helpful suggestions. We address reviewer comments below

• We have reworked the formatting to match the files you kindly mentioned. 

• We noticed you have some minor occurrence of overlapping text with the following previous publication(s), which needs to be addressed:

https://link.springer.com/article/10.1007%2Fs10943-019-00820-y

The text that needs to be addressed involves parts of the Introduction. In your revision ensure you cite all your sources (including your own works), and quote or rephrase any duplicated text outside the methods section. Further consideration is dependent on these concerns being addressed.

• We have revised the introduction to address the overlap you kindly mention. We cite our other publication early on when we revisit issues in literature we presented before. We paraphrased, quoted and changed some sentences to address this. We are happy to work with the editor if any specific further adjustments are needed.

• We note that you have indicated that data from this study are available upon request. PLOS only allows data to be available upon request if there are legal or ethical restrictions on sharing data publicly. In your revised cover letter, please address the following prompts:

o a) If there are ethical or legal restrictions on sharing a de-identified data set, please explain them in detail (e.g., data contain potentially identifying or sensitive patient information) and who has imposed them (e.g., an ethics committee). Please also provide contact information for a data access committee, ethics committee, or other institutional body to which data requests may be sent.

o b) If there are no restrictions, please upload the minimal anonymized data set necessary to replicate your study findings as either Supporting Information files or to a stable, public repository and provide us with the relevant URLs, DOIs, or accession numbers. Please see http://www.bmj.com/content/340/bmj.c181.long for guidelines on how to de-identify and prepare clinical data for publication. For a list of acceptable repositories, please see http://journals.plos.org/plosone/s/data-availability#loc-recommended-repositories.

We now address the information you kindly requested in the revised cover letter.

Reviewer 1:

• The paper is well written and described the instrument and the clinicians’ perspectives of it. To allow future use and comparison studies, please provide the Egyptian dialect version of Arabic as a supplementary material

Thank you for your suggestion. We have uploaded an Arabic version of the tool as you kindly requested.

Reviewer #2: 

• This is an excellent innovative study where 21 Muslim and non-Muslim clinicians provided valuable feedback on the implementability and use of RAMCOM, a new communication tool designed to assist in caring for fasting Muslim patients. I have revised the whole manuscript and RAMCOM tool (figure 1) and added my questions and comments in the attached word document. There are also few minor grammatical errors. Please send me the final draft after you answer my few questions and edit the manuscript for a final revision. Again, thank you for the well written and interesting article. I am sure it will have a great benefit to Muslim patients around the world.

Best regards,

Heba Abolaban, MD, MPH

Dear Dr Abolaban,

Thank you very much for your kind words and your meticulous examination of the paper. We addressed minor changes you kindly noted in the manuscript itself. For comments requiring further clarifications, we provide responses here.

• Lines 88 to 91: Rulings related to pregnant women have been revised

• Line 138: An explanation of what RAMCOM stands for has been provided.

• Lines 259 to 265: The sentence has been revised and an explanatory quote has been added.

• Lines 325 to 326: The sentence has been revised for better clarity.

• An emphasis on cultural competency training has been added throughout the manuscript as you kindly recommended. 

• The first question of RAMCOM now reads “What would you would like me to know about your faith that could help me provide the best health care to you?”

• Lines 355 to 358: The incorporation of RAMCOM within the AIDET framework has been added. Thank you for sharing your experience and for this suggestion!

• The conclusion has been revised so that recommendations are more specific.

• We revised the acknowledgement section to include compensation for clinicians. The phase addressing conflict of interest has been added in the submission system earlier. Competing interests: The authors have declared that no competing interests exist. Our understanding is that it appears on top in all manuscripts.

Reviewer #3: 

• Thank you for the opportunity to review this manuscript exploring an important healthcare need affecting Muslim patients.

• Thank you for your kind words and for noting the importance of this work. We appreciate your thoughtful remarks and have incorporated them as you would kindly see below. 

• Please be sure the manuscript adheres to & cites the appropriate Equator Network guidelines. For example: O'Brien BC, Harris IB, Beckman TJ, Reed DA, Cook DA. Standards for reporting qualitative research: a synthesis of recommendations. Acad Med. 2014;89(9):1245-1251.

Thank you for your suggestion. Kindly see the table below describing how address criteria mentioned in the O’Brien paper below. We cite the guidelines on lines 133-134

Topic How it was addressed in article

Title We revised the title to reflect this

Abstract Abstract meets criteria in agreement with PLOS guidelines

Introduction 

Problem formulation Kindly see revisions below.

Purpose or research question The research question has been revised as noted below.

Methods 

Qualitative approach and research paradigm Qualitative approach : A description of qualitative content analysis and how it was used is provided in the methods section

Guiding theory if appropriate: The theory of planned behavior is provided in the analysis section

Rationale: Provided under ‘Interview method and research instrument’

Researcher characteristics and reflexivity, Sampling strategy and techniques to enhance trustworthiness Details of reflexivity, sample selection, recruitment and strategies addressing trustworthiness as defined by Lincoln and Guba have been described in the first manuscript published from this project. This is mentioned and cited in the methods section.

Context Setting, site and other contextual factors are described in the methods section

Ethical issues pertaining to human subjects Ethical approval is provided in the methods section including the protocol number.

Units of study, Data collection methods ,instruments and technologies Provided under ‘Interview method and research instrument’

Data processing Methods for processing data prior to and during analysis, including transcription, data management , data coding are provided

Data analysis A separate subsection addresses this

Results 

Synthesis and interpretation Findings are explained in the context of the theory of planned behavior and GUIDE-IT model

Links to empirical data Evidence: Quotes were used to substantiate analytic findings

Discussion 

Integration with prior work, implications,

transferability, and contribution(s) to the field Discussion starts with a short summary of main findings; explanation of how findings and conclusions connect to, support, elaborate on, or challenge conclusions of earlier scholarship. Also provided is a discussion of context of application and identification of contribution(s) to scholarship in a discipline or field

Limitations Strengths and limitations are described in a separate subsection

Conflicts of interest and funding Provided in the submission system 

• Title: Please provide a concise description of the nature and topic of the study identifying the study as qualitative or indicating the approach (e.g., ethnography, grounded theory) or data collection methods (e.g., interview, focus group)

• We revised the title to reflect this. 

• Please clarify any hypothesis tested.

• We adhere to the school of thought that considers a research question rather than a hypothesis (i.e., prediction that involves variables and statistical tests) as the most suitable way of addressing qualitative research. 

Creswell, J. W. (2014). Research Design: Qualitative, Quantitative and Mixed Methods Approaches (4th ed.). Thousand Oaks, CA: Sage. Page 139

We do note, however, that the research question should have been mentioned more explicitly and clearly. We revised the research question in the abstract and the introduction to address this as you would kindly see below. 

This second manuscript complements earlier findings by proposing a communication tool, RAMCOM, which aims to assist clinicians in communicating with Muslim patients considering fasting during Ramadan. Using a theory driven qualitative study design, this study addresses the following questions:

What are the clinicians’ preferences for the tool in terms of Content, Format, Style, Length and language? How do clinicians perceive factors that would impact their intentions to sue the tool? What would facilitate the implementation and dissemination of the proposed tool?

• INTRODUCTION

Please provide a more referenced review of relevant theory and empirical work. This may include references to the Qu’ran, Sunnah (in form of Hadith), or ijtihad (independent legal reasoning), istihsan (preferential reasoning of jurists), al-urf (local customary precedent), and al-masalih al-mursalah (public interest or welfare) among others. There is a juristic heirarchy and these do not all rise to the same levels of precedent. 

• Thank you for pointing this out. We have revised our introduction to make sure specific references are used throughout.

• Also consider making the distinction between Halal (permissible or lawful), Haram (prohibited), and Makruh (discouraged but not legally forbidden). Also important to note is the idea that under Islamic law “ijtihad is not reversible” (al-ijtihad la yunqad), meaning that one ruling of ijtihad is not reversed by another of differing opinions. Disagreements (ikhtilaf) among jurists are seen in a positive light; legal texts record different juristic opinions on the same issue with a specific line of literature devoted to disagreements between jurists (ikhtilaf al-fuqaha). This juristic ikhtilaf is key to understanding the development of the Islamic legal tradition, and can provide an important juristic tool to interpreting Sharīʿah Law as it pertains to health and medicine. Many find this idea confusing. Indeed, classical Muslim scholars have reminded us that: our opinion is a right one with the possibility of being wrong and others’ opinions are wrong ones with the possibility of being right.

• We address this in the introduction as you kindly noted. Please see lines 77 and 72 to 75

We tried to keep this focused so that this technical information, while very useful, would not detract the readers (mostly health professionals) from key goals for the study. 

• Line 59-60: Consider further clarifying the root of this obligation. For example: It is a religious obligation for able-bodied adult Muslims as prescribed in the Qur’an 2:186. Fasting is obligatory upon every adult Muslim who is of sound mind, not ill, and not travelling > 80 km from city of residence (Quran 2:184).

• Thank you. We clarify this issue and cite the Qur’ an. Lines 61 to 63

• Line 61-62: Intent plays an important role in determining if an action invalidates ones fast. Consider clarifying this. For example: Many things that come into or out of the body invalidate fasting, such as intentional eating or drinking, oral medications that reach the stomach, deliberate vomiting, parenteral nutrition, and others.

• Thank you for the suggestion. We address this on lines 76 to 81

• Lines 65-58: Please provide more references on these topics as they are important and there are numerous fatawat on the topic; I know of at least 13 (1 Sunni, 11 Shi’a, 1 joint Sunni and Shi’a). When citing any fatawa, please be sure to specify if it is Sunni or Shi’a, and the Madhab if known.

• We made sure the source of every fatwa we cite is described. 

Lines 66-72 and 88-89 We provide a reference to recommendations of the 9th Fiqh-Medical seminar, which came from a symposium that gathered Islamic jurists with different opinions on dosage forms and fasting in 1997, while noting here and at other occasions that multiple opinions continue to exist. Again, while noting the importance of this, we try to make sure the introduction remains focused and tailored to the intended reader. 

• Line 72-74: This may be justified by Qur’an 2:195: …and let not your own hands throw you into destruction.

• Thank you! A reference to the Qur’an has been added to justify the statement. Lines 94-96

• Lines 182-185: Might consider incorporating emergency medicine physicians on future studies, as they may play a unique role when evaluating patients during more acute illnesses.

• This is a very good suggestion. We have included it on lines 392-393

• RESULTS / DISCUSSION

Thank you for including a copy of the tool. One might consider compiling a list of available evidence sorted by sect and/or madhab addressing this topic that could be listed on the back side of the page for further reference for patients or providers.

• Thank you for this helpful suggestion. We have included a recommendation of compiling such a list to assist clinicians if they and their facilities choose to do so. Lines 358 to 361

---

## [Editor Report · Decision Letter 1]

27 Jan 2020

RAMCOM: A qualitative study of clinicians’ viewpoints on a tool for communication with Muslim patients considering fasting during Ramadan

PONE-D-19-30447R1

Dear Dr. Amin,

We are pleased to inform you that your manuscript has been judged scientifically suitable for publication and will be formally accepted for publication once it complies with all outstanding technical requirements.

With kind regards,

Andrew Carl Miller

Academic Editor

PLOS ONE

Additional Editor Comments (optional):

Thank you for submitting this important and timely manuscript. You have satisfactorily addressed the reviewer's queries.
---

## [Editor Report · Acceptance letter]

29 Jan 2020

PONE-D-19-30447R1 

RAMCOM: A qualitative study of clinicians’ viewpoints on a tool for communication with Muslim patients considering fasting during Ramadan 

Dear Dr. Amin:

I am pleased to inform you that your manuscript has been deemed suitable for publication in PLOS ONE. Congratulations! Your manuscript is now with our production department. 

With kind regards,

on behalf of

Dr. Andrew Carl Miller 

Academic Editor

PLOS ONE